# Association between early child development trajectories and adult cognitive function in a 50-year longitudinal study in Guatemala

Ines Gonzalez Casanova ,[1,2] Ann DiGirolamo,[3] Maria F Kroker-Lobos,[4] Laura Ochaeta,[4] Manuel Ramirez-Zea,[4] Reynaldo Martorell,[2] Aryeh D Stein[2]

[1]Department of Applied Health Science, Indiana University Bloomington School of Public Health, Bloomington, Indiana, USA
[2]Hubert Department of Global Health, Rollins School of Public Health, Emory University, Atlanta, Georgia, USA
[3]Georgia Health Policy Center, Andrew Young School of Policy Studies, Georgia State University, Atlanta, Georgia, USA
[4]INCAP Research Center for the Prevention of Chronic Diseases, Institute of Nutrition of Central America and Panama, Guatemala, Guatemala

**Correspondence to**
Dr Aryeh D Stein;
aryeh.stein@emory.edu

## ABSTRACT

**Objectives** Few studies have used longitudinal data to study the development of cognition over the life course in low-income and middle-income countries. The objectives of this study were to assess predictors of cognitive development trajectories from 6 months through 7 years, and if these trajectories predicted adult cognitive function in a birth cohort from Guatemala.

**Design** We analysed data from the INCAP Nutrition Supplementation Trial Longitudinal Study in Guatemala. Cognition was assessed at eight different time points between 6 months and 7 years. We derived childhood development trajectories using latent class growth analysis. We assessed predictors of the trajectories using ordinal logistic regression, and associations between childhood trajectories and adult non-verbal intelligence and literacy at age 18–52 years (mean±SD =42.7±6.4 years) using mixed models.

**Setting** The study was conducted in four Guatemalan villages.

**Participants** The study included 927 participants from Guatemala with repeated measurements of cognitive function during the first 7 years of life.

**Results** We identified three trajectories of cognitive development (high: n=214, average: n=583, low: n=130). Participants whose mothers were taller (proportional log odds (PO)=0.03, 95% CI=0.01 to 0.06), had more years of schooling (PO=0.15, 95% CI=0.06 to 0.25), or lived in households with higher socioeconomic scores (PO=0.19, 95% CI=0.09 to 0.29) were more likely to follow higher trajectories. Childhood trajectories predicted adult non-verbal intelligence (high=18.4±0.3, average=14.6±0.53, low=11.4±0.9) and literacy (high=63.8±2.0, average=48.6±1.2, low=33.9± 2.6) scores.

**Conclusions** In this sample from Guatemala, cognitive development trajectories from 6 months through 7 years were associated with adult non-verbal intelligence and literacy. These findings provide evidence of tracking of cognition over time in a transitioning low-income setting.

## INTRODUCTION

In low-income and middle-income countries (LMICs), 43% of children younger than 5 years were considered at risk of poor development in 2010 using stunting and poverty as

### Strengths and limitations of this study

► Repeated measures of cognitive function from 6 months to 55 years of age.
► Birth cohort with 50 years of follow-up from Guatemala, a transitioning country.
► Longitudinal analysis using latent class growth analysis.
► There was missing information from some participants.
► Results are consistent with what has been reported in studies from high-income countries.

proxies.[1] Early childhood development and childhood intellectual functioning predict survival, lifelong health and human capital[2–5]; hence, assuring that children worldwide reach their full cognitive developmental potential is central to the sustainable development agenda.[1]

This has led to a widespread call to implement effective interventions that improve early cognitive development.[1 6 7] However, the complex nature of the known predictors of early development and challenges assessing development and intelligence across time in diverse populations have complicated the targeting and evaluation of interventions to improve life-course cognitive functioning, especially in LMICs.[8 9]

Understanding how cognitive functioning tracks across time in diverse settings is an important step towards identifying effective approaches to improve childhood development; however, information from LMICs is limited. Considering that LMICs have the greatest proportion of children at risk and are primary targets for interventions and that the first years of life are the most important period for brain growth and differentiation, it is important to address these gaps in knowledge from LMICs. Following a life-course

approach and taking advantage of a long-running cohort from Guatemala,[10] the objective of this study was to assess childhood cognitive developmental trajectories during the first 7 years of life, to identify predictors of these trajectories, and to estimate the associations of these trajectories with adult cognitive functioning.

## METHODS
### Longitudinal study
We used data from the INCAP Nutrition Supplementation Trial Longitudinal Study.[10] The study was originally conceived as a trial to test the hypothesis that providing a protein-energy supplement during the preschool years would improve mental development. The specific methods for the original trial have been described extensively[10–13]; briefly, four villages in Guatemala were assigned to receive either a protein-energy gruel (Atole; 119–163 kcal per serving) or a protein-free sugary drink (Fresco; 59 kcal per serving). The supplements were provided to all residents in the study villages in a central location twice daily from 1969 through 1977; intakes were recorded for pregnant and lactating women and children under the age of 7 years. In total, 2392 children were enrolled in the original study because they were either younger than 7 years at the beginning of the trial or were born during its implementation; hence, participants in this cohort were born between 1962 and 1977 in the study villages.[14] Follow-up studies have been conducted in 1988–1999, 2002–2004, and 2015–2017.[10 15 16]

### Participant and public involvement
Participants and community were not involved in the initial design of the intervention or outcome selection, but over the years the study team provided feedback to the communities about the study and its findings.

### Analytical sample and data for this analysis
We included all participants with at least three measures of cognitive development out of the eight potentially available. This is a conservative approach as latent class growth analysis can handle some participants having as few as one measurement.[17] For the analysis testing associations with adult intelligence and literacy, we excluded participants missing information on these measurements or those without measurements after 18 years of age. Information on sociodemographic predictors such as parents' years of schooling, household socioeconomic status and distance to the health centre was extracted from data collected during the original supplementation study (between 1969 and 1977).

### Measures
### Cognitive development assessments
The cognitive function assessments for infants and preschool children were specifically developed for the study using items from different age-appropriate tests.[18]

### Infant battery (administered at child ages 6, 15 and 24 months)
This set of instruments was administered within a 2-week window of infants turning 6, 15 and 24 months. It was developed as a compendium of relevant items from validated infant scales including the Bayley Scales of Infant Development,[19] the Gesell and Amatruda's Developmental Diagnosis,[20] the Cattell Infant's Intelligence Scales[21] and Merrill-Palmer Standards of Physical and Mental Growth.[22] The selected items were translated into Spanish and adapted for the Guatemalan context by trained psychologists and psychiatrists. The Composite Infant Score (CIS) summarises 44 items derived from the previously mentioned scales.

### Preschool battery (administered at child ages 3, 4, 5, 6 and 7 years)
A set of 22 tests was applied within a month of the participants' fifth, sixth and seventh birthday, and a shorter set of 10 tests was applied within a month of children's third and fourth birthdays. The preschool battery measured various dimensions of cognition (eg, verbal ability, general knowledge, quantitative skills and memory). These scales have previously been analysed by factor analysis, resulting in a general and a memory factor, which were deemed reliable and valid predictors of later development.[23]

### Raven's progressive matrices (RPM)
The RPM was applied during the 1988–1989, 2004–2005 and 2015–2017 data collection waves, when participants were between 11–27, 22–43 and 37–55 years old, respectively. The RPM is a non-verbal test of intellectual functioning and abstract reasoning that includes a series of pattern-matching exercises that get progressively more complex.[24] The RPM traditionally consists of five sets of 12 items each, however only the first three sets were applied, because in our pilot work we found that few participants could process further than that.[14] The score was calculated by summing the number of correct answers. The RPM has good test–retest reliability (intraclass correlation coefficient, 0.87) and internal consistency, as well as construct validity in this Guatemalan population.[14]

### Adult literacy and reading comprehension
### InterAmerican Reading Series (Serie Interamericana)
This test was administered in 2004–2005 and 2015–2017. To complete this test, those who were unable to read a test sentence from the local paper or had not completed primary school, received the prescreening test questions where scores could range between 0 and 5. Those who passed the screening test (by answering the five questions correctly) or had at least a secondary school education started with a score of 5 which was then added to the InterAmerican reading and writing series composite score. This approach has been found to have adequate test–retest reliability and internal consistency in this population.[25]

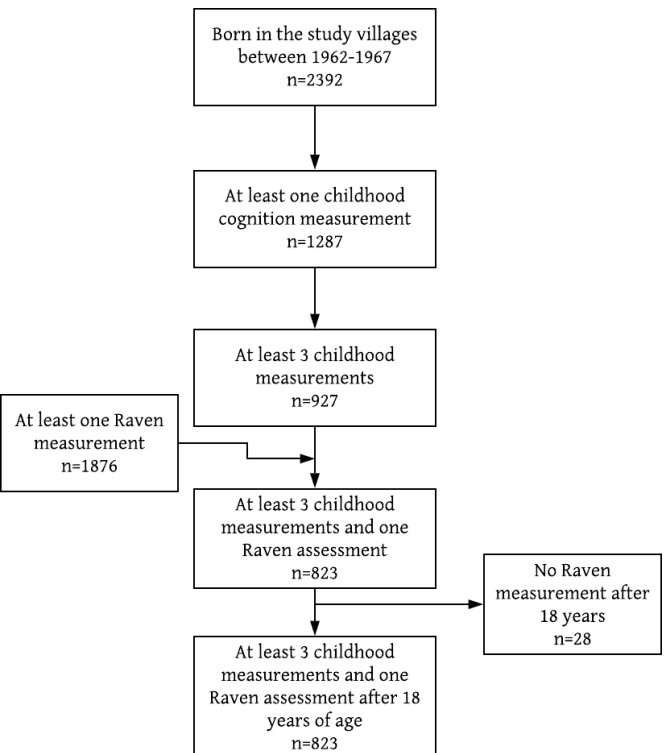

**Figure 1** Sample flowchart for cognitive development trajectory analysis in the INCAP Nutrition Supplementation Trial Longitudinal Study in Guatemala.

## Statistical analysis

### Summarising child cognitive data into trajectories

In order to make the scores comparable over time, we first converted the individual scores for mental development on the CIS at 6, 15 and 24 months, and the preschool series composite score at 3, 4, 5, 6 and 7 years into z-scores relative to the same population and age. We assessed Spearman pairwise correlations between all the available childhood cognitive development measurements during the first 7 years and RPM measurements stratified by age. We then used Latent Class Growth Analysis (LCGA)[17] to cluster participants into developmental trajectories during the first 7 years based on the age-adjusted ranked scores. LCGA classifies individuals based on their cognitive development at different ages into classes or trajectories with distinct intercepts and slopes, thereby capturing the heterogeneity within a sample. Because of the way the original study was designed, in order to account for potentially systematically missing information by birth year (online supplemental file 1), a sensitivity analysis was conducted by including children with at least one CIS and at least one preschool measurement. All LCGA were adjusted for sex and for maternal ID (to account for siblings in the sample). The LCGA analysis was performed using M-Plus (V.7.0, Los Angeles, CA).

### Assessing predictors of child cognitive development trajectories

We assessed if family-level variables predicted the trajectories using ordinal logistic regression. Models were developed based on theory and available data on the household sociodemographic determinants of child development, which included socioeconomic status (SES) score, parents' education, parents' education and height, family size (number of people living in the same household) and distance from the supplementation centre. All models were adjusted for type of supplement (operationalised as village of residence) and age at exposure to the supplement (first 1000 days versus not within the first 1000 days). Multiple imputation in SAS V.9.4 was conducted to account for item-specific missing values in the predictors. We used PROC MI to generate 20 imputed datasets using fully conditional specification. Ordinal logistic regression was then conducted for each of the 20 imputed datasets and the estimates were pooled using PROC MIANALYZE.

### Testing associations between child development measurements, trajectories and adult outcomes

We conducted a mixed effects linear model using PROC MIXED in SAS to test differences in the intercept of all available repeated measurements of adult RPM scores by cognitive development trajectory and by 7-year tertile. For the literacy score, we used the last score recorded and conducted analysis of variance tests to assess differences by trajectory and by 7-year tertiles as comparison. Only participants with measurements after 18 years of age were included in these analyses. We tested differences in sociodemographic predictors between included sample and those that had cognitive trajectory information but were lost to follow-up due to missing adult measurements.

Data management and analysis were conducted using SAS V.9.4 (SAS Institute Inc, Cary, North C, USA) unless otherwise noted.

## RESULTS

The sample for the trajectory analysis and early predictors included 927 participants with at least three measurements of cognitive development during childhood (online supplemental file 1). Sample sizes for testing associations between trajectory membership and adult cognitive functioning varied; in total, 823 participants could be classified into childhood trajectories and had at least one RPM measurement. Twenty-eight participants had their last RPM measurement conducted when they were younger than 18 years and were excluded from the analysis of trajectories and adult cognitive development. The final sample included for this analysis was of 795 participants (figure 1) and had similar sociodemographic characteristics to the sample lost to adult follow-up (n=132), except because a larger proportion of men were lost to follow-up (data not shown).

In the LCGA, based on model fit indexes, the sample membership for each trajectory and interpretability of the resulting classes, we identified three trajectories which we labelled low, average and high (figure 2). A sensitivity analysis including only children with at least one infant and one preschool measurement (n=577) also resulted in three similar trajectories. Differences among

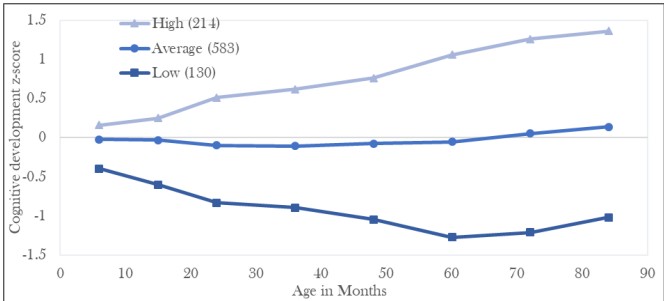

**Figure 2** Cognitive development trajectories from 6 months to 7 years (84 months) of age (n=927).

the three trajectories were apparent already at 6 months and became larger as the children grew older.

The weakest Spearman correlations with the adult RPM assessments were with the infant battery at 6 and 15 months and with the preschool battery at 3 years, followed by the infant battery at 2 years and the preschool battery at 4 years. All the correlations between the 5-year, 6-year and 7-year measurements, the trajectories and the adult RPM were in the moderate range. The strongest correlations between childhood assessments and the RPM were with the preschool battery at 6 years, which ranged between 0.40 (at 11–20 and 21–36 years) and 0.43 (at 37–55 years). The 7-year preschool battery had the strongest correlation with the RPM at ages 11–20 years; the trajectories had the weakest correlation with measurements at 11–20 years (0.30) but stronger correlations with the RPM at 21–36 years (0.35) and 37–55 years (0.38). The RPM at 11–20 years had a correlation of 0.54 with the RPM at 37–55 years (table 1).

Participants in the high trajectory belonged to households that, on average, had higher socioeconomic status, had mothers and fathers with more years of schooling, and had taller mothers, relative to those in the average trajectory. Conversely, the low trajectory had, on average, lower household socioeconomic status, their parents had fewer years of schooling and their mothers were shorter than those in the average trajectory (table 2).

In proportional odds cumulative logit models, maternal and paternal education and height, and SES were positively associated with the trajectories, while family size was inversely associated with the trajectories (table 3). After adjusting for all other covariates, the positive association with maternal schooling, maternal height and household SES remained, as did the inverse association with family size.

Child development trajectories were associated with adult RPM and literacy scores (table 4). The mean and SD for the individual average of RPM scores was 15.2 and 5.2, respectively; on average, participants in the low childhood cognitive development trajectory scored 7 fewer points on the RPM compared with those in the high trajectory (a difference of 1.3 SD of the mean). The mean and SD scores for the latest literacy scores were 49.8 and 28.6, respectively. The mean difference between the low and high trajectory for literacy scores was 29.9 (which represents 1.0 SD of the mean). The 7-year cognitive

**Table 1** Spearman pairwise correlations between cognitive development tests from 6 months through adulthood (n=1484 with at least 2 measurements of intelligence)

| R(n)* | Infant battery | | | Preschool battery | | | | | Trajectories | Raven's progressive matrices | | |
|---|---|---|---|---|---|---|---|---|---|---|---|---|
| | 6 Months | 15 Months | 24 Months | 3 Years | 4 Years | 5 Years | 6 Years | 7 Years | | 11–20 Years | 21–36 Years | 37–55 Years |
| **Infant** | | | | | | | | | | | | |
| 6 Months | 1.00 (397) | 0.28 (312) | 0.14 (293) | 0.02 (344) | 0.03 (342) | 0.11 (277) | 0.03 (180) | −0.11 (88) | 0.14 (361) | 0.08 (266) | 0.11 (289) | 0.08 (201) |
| 15 Months | | 1.00 (469) | 0.28 (312) | 0.09 (425) | 0.02 (425) | 0.17 (332) | 0.05 (222) | 0.09 (129) | 0.21 (447) | 0.04 (325) | 0.14 (345) | 0.11 (236) |
| 24 Months | | | 1.00 (466) | 0.36 (431) | 0.09 (426) | 0.16 (361) | 0.28 (256) | 0.25 (165) | 0.39 (452) | 0.11 (296) | 0.21 (352) | 0.18 (238) |
| **Preschool** | | | | | | | | | | | | |
| 3 Years | | | | 1.00 (736) | 0.29 (692) | 0.29 (570) | 0.26 (444) | 0.24 (329) | 0.45 (691) | 0.03 (398) | 0.09 (567) | 0.03 (388) |
| 4 Years | | | | | 1.00 (860) | 0.41 (696) | 0.33 (567) | 0.31 (443) | 0.53 (806) | 0.13 (414) | 0.21 (656) | 0.23 (450) |
| 5 Years | | | | | | 1.00 (833) | 0.60 (664) | 0.54 (533) | 0.71 (793) | 0.35 (328) | 0.34 (596) | 0.32 (443) |
| 6 Years | | | | | | | 1.00 (791) | 0.66 (632) | 0.77 (663) | 0.43 (231) | 0.40 (502) | 0.40 (440) |
| 7 Years | | | | | | | | 1.00 (739) | 0.76 (540) | 0.45 (140) | 0.34 (409) | 0.35 (422) |
| **Trajectories†** | | | | | | | | | 1.00 (927) | 0.30 (409) | 0.36 (667) | 0.38 (495) |
| **Raven** | | | | | | | | | | | | |
| 11–20 Years | | | | | | | | | | 1.00 (1297) | 0.57 (753) | 0.54 (523) |
| 21–36 Years | | | | | | | | | | | 1.00 (1452) | 0.96 (855) |
| 37–55 Years | | | | | | | | | | | | 1.00 (1163) |

*Values are Spearman correlations between child z-scores and adult raw Raven scores.
†The trajectories were coded low=0, average=1 and high=2 for this analysis.

**Table 2** Sociodemographic and childhood characteristics by cognitive development trajectory

| | Low (130) | Average (583) | High (214)* | P value |
|---|---|---|---|---|
| *Community-level predictor* | | | | |
| Large village (%) | 56.1 | 55.3 | 55.0 | 0.97 |
| *Household-level predictors at enrollment* | | | | |
| Distance from supplementation centre (km) | 2.7±0.1 | 2.5±0.1 | 2.4±0.1 | 0.26 |
| SES score | −0.5±0.1 | −0.2±0.1 | 0.2±0.1 | <0.001 |
| Maternal schooling (years) | 1.0±0.1 | 1.2±0.1 | 1.8±1.1 | <0.001 |
| Paternal schooling (years) | 1.3±0.2 | 1.5±0.1 | 2.0±0.1 | <0.001 |
| Maternal height (cm) | 148.0±0.5 | 148.9±0.2 | 150.0±0.4 | <0.001 |
| Paternal height (cm) | 160.1±0.6 | 160.6±0.3 | 161.6±0.6 | 0.12 |
| Family size | 6.8±0.2 | 6.7±0.1 | 6.5±0.2 | 0.34 |
| *Individual-level predictors* | | | | |
| Male (%) | 52.6 | 51.6 | 54.2 | 0.83 |
| Height 36 months (cm) | 83.5±0.4 | 85.5±0.2 | 86.7±0.3 | <0.001 |
| Weight 36 months (kg) | 11.3±0.2 | 11.8±0.1 | 12.0±0.1 | <0.001 |

*Trajectory analysis was adjusted by sex and age at measurement. Differences among groups were assessed using $\chi^2$ test for categorical variables and analysis of variance for continuous variables.

development score tertiles were also associated with RPM and literacy scores in adulthood. Those in the highest 7-year tertile scored on average 5.9 points higher on the RPM and 25.9 points higher on the literacy test compared with those in the lowest tertile.

## DISCUSSION

The primary aim of this study was to use a life-course approach to assess child development longitudinally in a longstanding cohort from Guatemala, which has 50 years of follow-up, and, in line with this approach, to identify predictors of early development and correlations with adult non-verbal intelligence and literacy. To our knowledge, this is the first study to assess life-course measures of cognition from early life

through adulthood in a LMIC using a developmental trajectory approach.

We identified three trajectories of childhood cognitive development through age 7 years that diverge early and then remain parallel through the early school years. Consistent with this finding, early factors such as household SES and family size at birth, as well as maternal years of schooling and height were associated with trajectory membership, independently of other sociodemographic variables. These findings support early life as a critical period and highlight the importance of holistic interventions that address the complex interplay between sociodemographic factors, which ultimately can determine lifelong health advantages or vulnerabilities.[26] They also highlight the

**Table 3** Proportional odds logistic regression of sociodemographic predictors and cognitive development trajectories (n=927)*

| | Unadjusted | | Adjusted† | |
|---|---|---|---|---|
| | Estimate* | 95% CI | Estimate* | 95% CI |
| socioeconomic status score | 0.22 | 0.13 to 0.32 | 0.19 | 0.09 to 0.29 |
| Family size (No of people) | −0.06 | −0.13 to 0.00 | −0.08 | −0.16 to −0.01 |
| Distance from supplementation centre (km) | −0.09 | −0.21 to 0.03 | −0.03 | −0.16 to 0.09 |
| *Parent's schooling (years)* | | | | |
| Maternal | 0.2 | 0.12 to 0.29 | 0.15 | 0.06 to 0.25 |
| Paternal | 0.12 | 0.05 to 0.18 | 0.07 | 0.00 to 0.15 |
| *Parent's height (cm)* | | | | |
| Maternal | 0.04 | 0.02 to 0.07 | 0.03 | 0.01 to 0.06 |
| Paternal | 0.03 | 0.00 to 0.05 | 0.01 | −0.02 to 0.04 |

*Estimates represent the proportional log odds of following a higher trajectory per unit increase in the predictor. Multiple imputation was used for item-specific missing values in the covariates. Trajectories were coded as low=0, average=1, high=2.
†Adjusted for all other covariates and for supplementation with Atole in the first 1000 days (sex and age at measurement were adjusted during the latent class growth analysis and thus were not included in this model).

**Table 4** Associations between cognitive development trajectories, preschool series tertiles at 7 years and adult literacy and non-verbal intelligence

| | Trajectory* | | | | 7-Years scores† | | | |
|---|---|---|---|---|---|---|---|---|
| | n | Low | Average | High | n | Low | Medium | High |
| Last available literacy score‡, ¶ | 776 | 33.9±2.6 | 48.6±1.2 | 63.8±2.0 | 154 | 33.7±2.5 | 43.6±2.1 | 59.6±2.0 |
| Intercept of all available non-verbal intelligence scores§ | 795 | 11.4±0.9 | 14.6±0.53 | 18.4±0.3 | 154 | 11.8±1.2 | 15.0±0.9 | 17.7±0.8 |

*Values are means and standard errors, mean differences by trajectory were assessed using mixed models (all p values<0.001).
†Values are means and standard errors by preschool series 7-year score tertiles, mean differences by tertile were assessed using mixed models (all p values<0.001).
‡Measured by the InterAmerican Series.
§Measured by the Raven Progressive Matrices test; models were adjusted for age at RPM measurement.
¶Mean±SD age at last measurement was 40.7±8.8 years.
RPM, Raven's progressive matrices.

importance of investing in maternal education and nutrition within a context of a transitioning society to break the intergenerational transmission of inequality.

Results from this analysis, besides supporting early life as a critical period for the development of life-course cognition, also show a cumulative process where single time point measurements at older ages (5–7 years) have stronger correlations with adult outcomes when compared with measurements conducted during infancy or early childhood (<4 years). The trajectory approach also provided more weight to the information from measurements conducted later in childhood. Taken together, results from this study highlight the importance of child cognitive development as a potential pathway linking early exposures and adult outcomes. This is important for the design and assessment of interventions and programmes aiming to improve life-course development, health, human capital, as well as economic development.

We found mid-size correlations of 0.38 between cognitive development trajectories through 7 years and non-verbal intelligence measured after 37 years of age. The trajectories created using repeated measurements of cognitive development by different instruments performed similarly to a single time point measurement at age 7 years of cognitive functioning in predicting adult intelligence and literacy in this sample from Guatemala. The use of trajectories has the added advantage of maximising the sample size, which can decrease bias. The stability of the trajectories decreases the need to conduct multiple tests that increase the risk of chance findings. Similarly, there is no consensus on the best age to assess the impact of interventions on development or to predict later cognitive outcomes as this might vary by setting and population[1 27 28]; trajectories allow utilising all available information rather than relying on a single cross-sectional measurement. Another advantage of using LCGA was that we were able to interpret the trajectories and make qualitative inferences about the timing and processes of early cognition within subcategories of this sample. For example, as mentioned before, the early separation of the trajectories highlights the importance of timing interventions during the prenatal period or early life.

The analysis of the correlation among the trajectories, individual childhood measurements and adult outcomes also provided insights into the predictive value of individual time-measurements at different ages, which would not be evident with single measurements. In this instance, we showed higher correlations of individual child development measurements after 5 years with both the trajectories and adult non-verbal intelligence, when compared with earlier measurements. These results could be reflective of methodological challenges assessing development in younger children,[11 29] as well as of cumulative processes and pathways that eventually lead to adult cognition.

A limitation using trajectories is that it can be costly to collect repeated measurements of cognition, and in some cases single measurements can be the only alternative to assess childhood cognitive function. Similarly, missing data are a limitation of this study. Less than half of the participants of the original study had three or more measurements of cognitive development and, due to the study design, most participants have missing data either at younger or older childhood measurements. However, to account for this, we conducted a sensitivity analysis including only those participants with at least one measurement during infancy and at least one measurement during the preschool period and found similar trajectories. Another limitation with the study design was the large age-span of the cohort participants (ie, 11 years), which forced us to include large age groups for adult measurements and complicated the study of specific periods during adulthood. A limitation of the LCGA approach is that the researcher selects the number of classes. We used model fit statistics and class size to select the number of classes but the process is empirical and not necessarily exempt from bias. Finally, there are additional factors that could have contributed to child cognitive development that were not measured in this study; for example, information on parents' intelligence or early childhood care and stimulation, among others factors, were not available.

Global estimates of the number of children not meeting their full developmental potential are currently based on poverty and stunting.[1 30] Even though these two factors are associated with early childhood development, the use

of proxies to assess cognition is limiting,[9] and addressing this limitation is important in order to improve our ability to study the determinants of life-course cognition.[9] Results from this study contribute to further our understanding of cognitive functioning across the lifespan in LMICs. This cognitive development trajectory approach sets the basis for the study of life-course predictors of intelligence, which in turn can improve the design and targeting of interventions in diverse populations.

**Contributors** All authors contributed substantially to the conception of the work, acquisition, analysis and/or interpretation of the data as required by BMJ authorship requirements. IGC participated in the conceptualisation, conducted the analysis and drafted the original manuscript. ADG, MFK-L and LO participated in the conceptualisation, data collection and validation and analysis. MR-Z and RM participated in the design and implementation of the original study and the follow-ups. ADS participated in the conceptualisation of the follow-ups and the present analysis, supervised and guided the implementation of this study and the writing of the manuscript. All authors reviewed the manuscript critically for intellectual content. All authors agree to be accountable for all aspects of the work reported in this manuscript.

**Funding** This study was funded by the Bill and Melinda Gates Foundation (#OPP1164115). The funder was not involved in the design or interpretation of the results of this analysis. IGC is funded by the National Heart, Blood and Lung Institute awards HL137338-03S1 and HL126146-02.

**Competing interests** None declared.

**Patient consent for publication** Obtained.

**Ethics approval** This study was approved by Emory University's Institutional Review Board IRB00071041. Parents or legal guardians provided written informed consent for childhood data collection and participants provided written informed consent for each follow-up.

**Provenance and peer review** Not commissioned; externally peer reviewed.

**Data availability statement** Data are available upon reasonable written request to aryeh.stein@emory.edu.

**ORCID iD**
Ines Gonzalez Casanova http://orcid.org/0000-0001-5747-8636

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
