## [Reviewer comments · BMJ Open]

ARTICLE DETAILS

TITLE (PROVISIONAL)	Association between early child development trajectories and adult cognitive function in a 50-year longitudinal study in Guatemala
AUTHORS	Gonzalez Casanova, Ines; DiGirolamo, Ann; Kroker-Lobos, Maria; Ochaeta, Laura; Ramirez-Zea, Manuel; Martorell, Reynaldo; Stein, Aryeh D.

VERSION 1 – REVIEW

REVIEWER	Pratesi, Claudia Universidade de Brasilia, Health Sceinces
REVIEW RETURNED	28-Oct-2020

GENERAL COMMENTS	This is an interesting study, and I enjoyed reading it. I did not see any IRB, consent, or Helsinki declaration. Was there any? Please review punctuation; most periods are after the reference; however, some are before. How many calories were in the protein supplement as opposed to the control?
---

REVIEWER	Knopman, DS Mayo Clinic, Department of Neurology
REVIEW RETURNED	15-Nov-2020

GENERAL COMMENTS	This analysis draws on a unique cohort of persons in Guatemala, who as children took part in a nutritional research program in which cognitive and developmental assessments were performed between ages 6 mo and 7 years. A subset was available for reevaluation as adults. The authors report that the developmental trajectories as children were related to maternal wellbeing (height, schooling, SES). They then went on to show that the most favorable developmental trajectories were associated with better performance at ages 11-55 yrs (mean 42) on a widely used nonverbal measure of reasoning and a measure of literacy. The use of a developmental trajectory is a novel approach and while it resulted in the loss of about 300 participants (who had <3 measurements), it is probably more stable than a single measurement. There is no question that this is a novel study group and the long duration of follow-up into middle age is of great value. Information of this sort from low/middle income countries is rare. There are some major limitations however some of which are unfixable and some fixable. As fully disclosed by the authors in Figure 1, about 2/3s of the original cohort (823/1287) was included in the current analysis. It probably is not possible to evaluate those with one childhood exam who were lost to follow-up but the authors
--

	should describe the group lost to adult evaluation so that the reader can better understand the survivor bias. The title and many points in the manuscript state that the analysis relates to cognition in adulthood (mean age 42 yrs), but in fact the last measurement occurred in some unknown number of people who were under age 18 (because age 11 is given as lower end of the range of follow-up. If the reviewer is reading the methods correctly, this would have truncated the follow-up cohort at some adult age, perhaps 25 though the older the better for supporting the claim of the manuscript. More details on the ages at which “adult” measurements were taken might have clarified this issue. The use of a developmental trajectory is a strength from an analytic viewpoint, but it perhaps needs to be clarified that the costs of using this methodology as an outcome in future prevention trials on a large scale for improving childhood might be too high. Can the authors propose a simpler strategy (eg just looking at age 5) for example?
--	---

REVIEWER	Rosner, Bernard Harvard Medical School, Biostatistics
REVIEW RETURNED	16-Feb-2021

GENERAL COMMENTS	Comments for the Author 1. Pg. 7, para. 2 Why only use the last RPM in the analyses? This is wasting 2/3 of the data collected. Instead, mixed effects linear regression can be used with individual RPM scores as repeated measures and controlling for age of administration of the RPM, early childhood trajectories and other covariates. 2. Pg. 9, para. 2 Are the trajectories still significant predictors after adjusting for the 7 year cognitive scores? 3. General question What was the effect of the nutritional supplements on adult cognitive scores? 4. Pg. 11, para. 1 Why focus on the prenatal period? It was only very weakly associated with adult RPM scores. 5. General question What was the effect of (a) the average childhood score and (b) the change in the childhood score over time in the preschool battery on adult RPM scores?
--

VERSION 1 – AUTHOR RESPONSE

Reviewer 1: I did not see any IRB, consent, or Helsinki declaration. Was there any?

We added a statement in the methods.

Please review punctuation; most periods are after the reference; however, some are before.

We revised punctuation for consistency.

How many calories were in the protein supplement as opposed to the control?

We now note that the atole supplement had 119 kcal per 180 ml serving for the first 4 months of life and 163 kcal per serving for older participants (4 mo-7 years). The control supplement (fresco) had 59 kcal per serving.

Reviewer 2:

As fully disclosed by the authors in Figure 1, about 2/3s of the original cohort (823/1287) was included in the current analysis. It probably is not possible to evaluate those with one childhood exam who were lost to follow-up but the authors should describe the group lost to adult evaluation so that the reader can better understand the survivor bias.

We conducted this analysis and found that the only significant difference was in the number of male participants who do not have adult measurements. We added a sentence to the methods and a sentence to the results stating this.

The title and many points in the manuscript state that the analysis relates to cognition in adulthood (mean age 42 yrs), but in fact the last measurement occurred in some unknown number of people who were under age 18 (because age 11 is given as lower end of the range of follow-up. If the reviewer is reading the methods correctly, this would have truncated the follow-up cohort at some adult age, perhaps 25 though the older the better for supporting the claim of the manuscript. More details on the ages at which “adult” measurements were taken might have clarified this issue.

We appreciate the potential for confusion. We excluded participants with measurements under 18 years of age and updated the results. There were 28 participants with trajectory data and where the latest Raven measurement was conducted when they were younger than 18 years. The results were updated based on this and the additional analyses requested by reviewer 3.

The use of a developmental trajectory is a strength from an analytic viewpoint, but it perhaps needs to be clarified that the costs of using this methodology as an outcome in future prevention trials on a large scale for improving childhood might be too high. Can the authors propose a simpler strategy (eg just looking at age 5) for example?

We clarified this point in the discussion.

Reviewer 3

Why only use the last RPM in the analyses? This is wasting 2/3 of the data collected. Instead, mixed effects linear regression can be used with individual RPM scores as repeated measures and controlling for age of administration of the RPM, early childhood trajectories and other covariates. We conducted the suggested analysis adjusted for age at administration of the RPM and updated the results.

Are the trajectories still significant predictors after adjusting for the 7 year cognitive scores?

Trajectories and age-7 cognitive scores are highly correlated and shared variance is a methodological concern with interpretation of models with both variables included. Nevertheless, neither the trajectories ($p=0.75$) nor the 7-year cognitive scores ($p=0.06$) are significant in models where both variables are included, but the coefficients are positive for both variables. The subsample with both trajectories and 7-year scores includes only 154 participants (output below).

Solution for Fixed Effects

Effect	traj	Estimate	Standard	Error	DF	t Value	Pr > t
Intercept		13.8241	2.1450	66		6.44	<.0001
traj 1		0.3854	2.2910	66		0.17	0.8669
traj 2		1.6629	3.1174	66		0.53	0.5955
traj 0		0					
cogz8		1.9586	1.0358	66		1.89	0.0630

Type 3 Tests of Fixed Effects

Effect	Num	DF	Den	DF	F Value	Pr > F
traj 2	66	0.28	66		0.7543	
cogz8	1	66	3.58	66	0.0630	

Least Squares Means

Effect	traj	Estimate	Standard	Error	DF	t Value	Pr > t
traj 1		14.9698	0.7863	66		19.04	<.0001

traj 2 16.2473 1.2923 66 12.57 <.0001
 traj 0 14.5844 2.3584 66 6.18 <.0001

What was the effect of the nutritional supplements on adult cognitive scores?

This has been published previously. Exposure to Atole in the first three years of life (relative to fresco) was associated with an increase of 0.3 (0.1) in reading comprehension z-score and of 0.2 in Raven z-score at mean age 33 y.

John A. Maluccio, John Hoddinott, Jere R. Behrman, Reynaldo Martorell, Agnes R. Quisumbing, Aryeh D. Stein, The Impact of Improving Nutrition During Early Childhood on Education among Guatemalan Adults, The Economic Journal, Volume 119, Issue 537, 1 April 2009, Pages 734–763, <https://doi-org.proxyiu.uits.iu.edu/10.1111/j.1468-0297.2009.02220.x>

Pg. 11, para. 1

Why focus on the prenatal period? It was only very weakly associated with adult RPM scores.

We are suggesting that interventions focus on the prenatal period or early childhood because the trajectories seem to be established early in life, hence interventions during this period are likely to make a significant impact in life-course cognition.

What was the effect of (a) the average childhood score and (b) the change in the childhood score over time in the preschool battery on adult RPM scores?

a) The average childhood score (including all measurements from 6 months to 7 years) was inversely associated with the intercept of adult raven tests (output below).

Solution for Fixed Effects

Effect Estimate Standard

Error DF t Value Pr > |t|

Intercept 14.9859 0.3858 163 38.84 <.0001

childx -1.3867 0.2955 163 -4.69 <.0001

b) Per each standard deviation monthly increase in the preschool battery score, on average participants had 0.03 (0.007) higher adult RPM score (output from mixed model below).

Solution for Fixed Effects

Effect Estimate Standard

Error DF t Value Pr > |t|

Intercept 17.0756 0.3526 793 48.43 <.0001

prescmix1 -0.1164 0.3931 2182 -0.30 0.7671

agemix1 -0.01864 0.006755 2182 -2.76 0.0058

prescmix1*agemix1 0.02897 0.007346 2182 3.94 <.0001

VERSION 2 – REVIEW

REVIEWER	Knopman, DS Mayo Clinic, Department of Neurology
REVIEW RETURNED	24-Mar-2021

GENERAL COMMENTS	in this revised version, the changes in text were acceptable to me. I was not able to review the author response letter as it was not available
---

REVIEWER	Rosner, Bernard Harvard Medical School, Biostatistics
REVIEW RETURNED	05-Apr-2021

GENERAL COMMENTS	1. Pg. 7, line 14
-------------------

	Why was only the last available RPM measurement used? 2. Table 2 Is the mean number of years of schooling really between 1.0 and 2.0 years/
--	---

VERSION 2 – AUTHOR RESPONSE

REVISION 2

We have addressed the additional comments as follows:

- As noted by one of the reviewers, it appears that your point-by-point response to the previous set of reviewers' comments was not provided with your revised manuscript. Please ensure to attach a point-by-point response with your next revision, and that you include responses to BOTH sets of comments, from this round of review and the previous one.

We have included both point-by-point responses and uploaded them with the manuscript files.

- Please also go through the previous reviewers reports and ensure that ALL points are addressed with appropriate changes to the manuscript. For example, you don't seem to have addressed Reviewer 1's comment "How many calories were in the protein supplement as opposed to the control?"

We made sure to address all points raised by the reviewers including the question about the amount of calories in the supplement and the control.

- Please ensure that your abstract is formatted according to our Instructions for Authors:
http://bmjopen.bmj.com/pages/authors/#research_articles

We formatted the abstract based on the guidelines provided.

- Many thanks for adding an Ethics statement to the Methods. In the statement, please specify if the consent obtained from parents/legal guardians and participants was written or verbal. If verbal, please include an explanation as of why no written consent was obtained and confirm if this was approved by the Ethics committee.

We have clarified that written consent was obtained.

In this revised version, the changes in text were acceptable to me. I was not able to review the author response letter as it was not available.

For the last submission, we uploaded the response to reviewers under "comments for reviewers" as an attachment. This time we have included it with the main manuscript files.

1. Pg. 7, line 14

Why was only the last available RPM measurement used?

Based on the reviewer's feedback, we updated the analysis to include all available repeated measurements of RPM and tested differences in the intercept of these measurements by trajectory. We have updated the text to clarify this.

2. Table 2

Is the mean number of years of schooling really between 1.0 and 2.0 years?

Yes, the histograms for years of schooling are provided below. The parents of the participants grew up in rural Guatemala in the 40s-50s and these schooling outcomes are in line with what would be expected for that context.